# Baseline and Dynamic Neutrophil-to-Lymphocyte Ratio as Prognostic Biomarkers in Metastatic Colorectal Cancer Patients Treated with Panitumumab

**DOI:** 10.3390/jpm16010009

**Published:** 2025-12-31

**Authors:** Teodor Marian Vancea, Andrada Olivia Tapirdea, Mihai Gabriel Zait, Daniel Sur, Calin Cainap, Claudia Burz

**Affiliations:** 1Faculty of Medicine, “Iuliu Hatieganu” University of Medicine and Pharmacy, 400349 Cluj-Napoca, Romania; tapirdea.andrada.olivia@elearn.umfcluj.ro (A.O.T.); zait.mihai.gabriel@elearn.umfcluj.ro (M.G.Z.); 2Department of Medical Oncology, The Oncology Institute “Prof. Dr. Ion Chiricuta”, 400015 Cluj-Napoca, Romania; 3Department of Medical Oncology, “Iuliu Hatieganu” University of Medicine and Pharmacy, 400012 Cluj-Napoca, Romania; 4Department of Immunology and Allergology, “Iuliu Hatieganu” University of Medicine and Pharmacy, 400162 Cluj-Napoca, Romania

**Keywords:** metastatic colorectal cancer, panitumumab, neutrophil-to-lymphocyte ratio, prognostic biomarker, progression-free survival

## Abstract

**Background**: Colorectal cancer (CRC) remains a leading cause of cancer mortality worldwide, with epidermal growth factor receptor (EGFR) inhibitors improving outcomes in patients with metastatic CRC (mCRC) harboring KRAS wild-type tumors. Inflammation has emerged as a potential determinant of treatment efficacy, and the neutrophil-to-lymphocyte ratio (NLR) represents an accessible biomarker reflecting the balance between protumoral neutrophils and antitumor lymphocytes. **Methods**: We conducted a retrospective study of 44 patients with left-sided, KRAS wild-type mCRC treated at the Institute of Oncology “Prof. Dr. Ion Chiricuta” between 2015 and 2024 with first-line FOLFOX/FOLFIRI plus panitumumab. Baseline NLR (bNLR) and NLR after four treatment cycles were assessed, with progression-free survival (PFS) as the primary endpoint. **Results**: Receiver operating characteristic analysis identified an optimal bNLR cutoff of 3.06 (AUC = 0.78, *p* = 0.004). Patients with low bNLR (≤3.06) had significantly longer PFS than those with high bNLR (>3.06) (14 vs. 7 months, HR = 2.7, *p* = 0.002). Notably, patients with a positive ΔNLR (increase during therapy) also demonstrated superior PFS compared to those with a negative ΔNLR (18 vs. 11 months, HR = 0.47, *p* = 0.022). In multivariate analysis, only bNLR remained independently associated with PFS. **Conclusions**: These results suggest that both baseline and dynamic NLR may serve as low-cost prognostic biomarkers in mCRC patients treated with anti-EGFR therapy.

## 1. Introduction

Colorectal cancer (CRC) represents the third most common malignancy and the second leading cause of cancer-related mortality worldwide as of 2022, with a particularly high prevalence among males over the age of 50 [1].

The etiology of CRC is largely attributed to genetic predisposition and lifestyle factors—including Western dietary patterns, obesity, alcohol consumption, and smoking—which collectively contribute to the geographical heterogeneity observed in its global distribution, with higher and rising incidences in high-income countries [2]. Notably, there has been a recent increase in early-onset colorectal cancer (EOCRC), defined as diagnosis before the age of 50, a trend likely driven by enhanced detection through screening and early exposure of the gut microbiome to carcinogens [3]. Despite advances in therapeutic and preventive strategies, metastatic colorectal cancer (mCRC) continues to carry a high mortality rate, with a five-year survival of about 15% [4].

The epidermal growth factor receptor (EGFR) is a transmembrane protein belonging to the tyrosine kinase receptor family, which activates downstream signaling pathways to promote cellular growth and proliferation. Genetic alterations that lead to aberrant EGFR activation result in sustained signaling, thereby driving tumor growth, proliferation, and angiogenesis. This signaling is mediated in part by the mitogen-activated protein kinase (MAPK) pathway and is regulated by the rat sarcoma (RAS) protein family [5]. Assessment of RAS mutational status as a predictive biomarker is mandatory prior to initiating therapy [6]. Anti-EGFR agents, such as cetuximab and panitumumab, are monoclonal antibodies that inhibit the EGF-mediated signaling cascade, thereby suppressing protumoral processes. These agents have demonstrated significant improvements in overall survival (OS) and progression-free survival (PFS) in patients with metastatic colorectal cancer (mCRC) with microsatellite-stable (MSS)/proficient mismatch repair (pMMR) and KRAS wild-type (KRASwt) tumors, when used in combination with chemotherapy [7,8,9].

Despite accepted benefit, tumor heterogeneity, the emergence of acquired resistance mechanisms, and individual immunologic profiles may diminish the efficacy of anti-EGFR therapies [10]. The principles of personalized medicine advocate for the integration of predictive biomarkers to maximize therapeutic benefit while minimizing harm [11]. Current guidelines from the European Society of Medical Oncology (ESMO, 2025) and the American Society of Clinical Oncology (ASCO, 2022) recommend mandatory testing of RAS exon 2, 3, and 4 mutational status, while extended RAS, BRAF V600E, and mismatch repair (MSI/MMR) profiles are advised before commencing treatment to predict response to anti-EGFR agents [12,13]. Some eligible patients still fail to respond or develop resistance, indicating the presence of multiple resistance mechanisms. Consequently, a more comprehensive pretreatment biological assessment is necessary to prevent futile adverse effects and unnecessary costs.

The neutrophil-to-lymphocyte ratio (NLR) serves as an indicator of systemic inflammatory activity [14]. Elevated neutrophil counts reflect pro-tumoral processes, whereas lymphocytes play a central role in anti-tumor immunity; thus, the NLR highlights the balance of immune activation. Pathophysiologically, high neutrophil counts are thought to promote tumor growth and dissemination by releasing pro-angiogenic factors, remodeling the extracellular matrix, and suppressing cytotoxic lymphocyte activity. Tumor-associated neutrophils present two phenotypes: N1 with anti-tumoral properties, mainly described as mature cells predominant in early stages, and N2 as pro-tumoral immature cells predominant in later stages. They are actively recruited in the tumor microenvironment (TME) by cytokines (IL-17, TNF, IGF) and chemokines. Inside the TME, neutrophils can be converted from N1 to N2 [15]. Hence, their main actions are to promote local invasiveness by regulating inflammatory processes and secreting pro-angiogenic factors (vascular endothelial growth factor) and metastatic proliferation through the formation of neutrophil extracellular traps, which capture circulating tumor cells [16,17].

Lymphopenia, on the other hand, reflects impaired immune surveillance, reducing the host’s ability to mount effective anti-tumor responses. This process is complex and divided into cancer immune surveillance as a circulating, non-specific mechanism and tumor-infiltrating lymphocytes as a distinct population recruited through signaling pathways inside the TME [18]. Both have a main anti-tumoral action by releasing cytotoxic enzymes against cancer cells. In addition, cancer has the canonical properties of immune evasion and immunosuppression, and the inflammation inside the TME is mainly responsible for the elevated amount of T regulatory cells, tumor growth factor (TGF-β), and N2 populations, which inhibit the anti-tumoral function of the lymphocytes [19]. Deficient intratumoral immunocytes infiltrate is also associated with a greater immune evasion [20].

A high NLR is indicative of predominant pro-tumoral inflammation [21]. Since 2005, randomized trials, retrospective analyses, and comprehensive reviews have established NLR as a predictor of poorer overall survival (OS) and progression-free survival (PFS) in colorectal cancer and other solid tumors [22,23,24].

The primary objective of our study was to correlate both baseline and early changes in NLR with PFS in a retrospective cohort of patients with mCRC treated with chemotherapy and anti-EGFR therapy.

## 2. Materials and Methods

### 2.1. Study Population

Patients with left-sided, KRASwt, metastatic colorectal cancer (mCRC) treated according to the ESMO guidelines with first-line FOLFOX/FOLFIRI (5 Fluorouracil + Oxaliplatin + Leucovorin/5 Fluorouracil + Irinotecan + Leucovorin) and Panitumumab, at the Institute of Oncology “Prof. Dr. Ion Chiricuta” Cluj-Napoca (IOCN), Romania, between January 2015 and December 2024, were enrolled in this retrospective cohort study. Patients with active infections (viral or bacterial), those who did not have the complete blood count available before treatment initiation, and those who did not undergo CT evaluation after 6 months were excluded from this study. Patients with prescribed antibiotic treatment, modified C-reactive protein, modified procalcitonin or positive for SARS-CoV-2 virus were excluded due to suspicion of active infections.

This study was conducted in accordance with the Declaration of Helsinki and was approved by the IOCN ethics committee (no. 342/08.08.2025). All patients have provided informed consent at the start of treatment. All data were obtained from the Institutional Cancer Registry.

### 2.2. Data Collection and Outcomes

Clinical data (complete blood count, CEA, CA-19.9) were collected for all patients at two different time points (before any treatment and after four cycles of therapy). Patient demographics and other clinical variables: age, sex, and metastasis site were also documented.

Tumor evaluation was carried out by computed tomography (CT) scans 4–6 months after treatment initiation. Treatment efficacy was quantified according to Response Evaluation Criteria in Solid Tumors version 1.1 (RECIST 1.1). Baseline NLR (bNLR) was defined as the ratio of absolute pre-treatment neutrophil count to absolute pre-treatment lymphocyte count. To evaluate NLR dynamics, the ratio after four cycles of therapy (tNLR) was calculated using the same method as for the baseline NLR (bNLR). The change in NLR (ΔNLR = tNLR − bNLR) was then analyzed to observe whether the ratio increased or decreased throughout treatment. bNLR was calculated before treatment initiation to capture the treatment-naive patient profile, while tNLR was calculated after four cycles of therapy as this interval aligns with the 8-to-12-week treatment assessment window recommended by ESMO guidelines.

Progression-free survival (PFS) was defined as the interval from treatment initiation until disease progression or death. Patients who still manifested disease control at the time of the last follow-up were censored. Objective response rate (ORR) was calculated as the proportion of patients who achieved complete response (CR) or partial response (PR). Disease control rate (DCR) was defined as the proportion of patients with CR, PR, or stable disease (SD).

### 2.3. Statistical Analysis

Receiver operating characteristic (ROC) curve analysis was performed for bNLR. The optimal cutoff value for predicting treatment efficacy was calculated using the Youden index (J = sensitivity + specificity − 1), and patients were divided into different groups. PFS was calculated and compared using the Kaplan-Meier method and the log-rank test. Univariate COX proportional risk regressions were used to identify independent predictors, and variables with *p* < 0.1 were included in a multivariate model. *p*-value < 0.05 was considered statistically significant. All of the statistical analyses were performed using IBM SPSS Statistics, version 20.0 (IBM Corp., Armonk, NY, USA).

## 3. Results

### 3.1. Cohort Characteristics

A total of 44 patients with mCRC treated with FOLFOX/FOLFIRI and panitumumab were included in the study. The median age at diagnosis was 59 years, ranging from 24 to 79. There were 23 (52%) males and 21 (48%) females. Among the patients, 24 (54%) presented with synchronous metastases and 20 (46%) with metachronous metastases. A total of 35 patients (80%) received FOLFOX, while 9 patients (20%) were treated with FOLFIRI. Liver metastases were observed in 31 patients (70%), lung metastases in 12 patients (27%), and peritoneal metastases in 10 patients (22%). Baseline clinical characteristics of the patients are summarized in Table 1.

### 3.2. Treatment Response and PFS

Treatment response, as evaluated by computed tomography (CT), showed that 16 patients (36%) achieved partial response (PR), 14 patients (32%) had stable disease (SD), 12 patients (27%) had progressive disease (PD), and 2 patients (4%) had complete response (CR). The median PFS was 11 months (95%CI: 8.9–13 months) and only three patients were censored (6.8%). The objective response rate (ORR) was 40% and the disease control rate (DCR) was 72%.

### 3.3. bNLR as a Predictor of DCR and PFS

At baseline, the median (range) level of bNLR was 2.71 (0.5–10.4). According to the ROC curve analysis (Figure 1), the optimal cut-off value in predicting DCR was 3.06. The sensitivity of this point was 75% and the specificity was 87.5%, with an AUC of 0.78 (95%CI: 0.59–0.98, *p*-value = 0.004).

According to the optimal cut-off value, patients were stratified into two groups, with 13 patients (30%) classified as high bNLR and 31 patients (70%) as low bNLR. Patients in the high bNLR group had a significantly worse PFS when compared to the low bNLR group [Median PFS: 7 months (95%CI: 5.2–8.7) vs. 14 months (95%CI: 9–18.9) *p* value = 0.002; HR = 2.7 (95%CI: 1.3–5.3)] (Figure 2).

### 3.4. Prognostic Value of NLR Early Changes for PFS

The median (range) level of NLR after four cycles of therapy was 1.81 (0.5–4.5). The median ΔNLR was −0.466 (range: −6.28 to 1.79). Based on the ΔNLR, the patients were divided into two groups (increase in NLR under treatment and decrease). Fifteen patients (34.1%) had a positive ΔNLR, and twenty-nine patients (65.9%) had a negative ΔNLR.

Moreover, we analyzed the impact of the alterations of NLR under treatment on the PFS. The results indicate that patients with a positive ΔNLR have a longer PFS compared with those with a negative ΔNLR [Median PFS: 18 months (95%CI:5.9–30) vs. 11 months (95%CI: 9–12.9) *p*-value = 0.022 HR = 0.469 (95%CI: 0.23–0.93)] (Figure 3).

### 3.5. Univariate and Multivariate COX Regression of PFS

Univariate and multivariate analyses of PFS were performed using COX regression. The variables considered included age, gender, CEA value, CA 19-9 value, timing of metastasis (synchronous or metachronous), bNLR, and ΔNLR. In univariate analysis, metachronous metastasis, bNLR < 3.06, and a positive ΔNLR were associated with longer PFS. In multivariate analysis, two independent COX regression models were established to avoid collinearity between bNLR and ΔNLR. Only bNLR < 3.06 was independently associated with longer PFS (Table 2).

## 4. Discussion

In this retrospective cohort of patients with left-sided, KRASwt mCRC treated with chemotherapy plus panitumumab, we found that both bNLR and its early changes were associated with clinical outcomes. A low bNLR (<3.06) predicted significantly longer PFS, in line with prior evidence supporting NLR as a prognostic marker in mCRC. More unexpectedly, we also observed that patients whose NLR increased after four cycles of therapy had superior PFS compared to those with stable or decreased values.

Notably, the median cut-off value of treatment-naive NLR (described in our study as bNLR), identified in a systematic review and meta-analysis conducted by Naszai et al. was 3.12 [25], closely aligning with the results observed in our cohort. The baseline findings are consistent with previous studies linking low bNLR to improved outcomes in patients receiving anti-EGFR therapy [26].

The dynamic results, however, are contradictory to most studies, wherein fixed or declining NLR was generally linked with a favorable outcome [27]. Liu et al. reported in a retrospective analysis that an early decline of NLR in patients with mCRC treated with FOLFOX was linked with a longer PFS [28]. The potential cause is the differing biology involved in EGFR inhibition. Panitumumab is capable of inducing inflammation and dermatologic toxicities associated with therapeutic efficacy [29]. The reversible neutrophil elevation seen in certain patients could, thus, be indicative of a process of immune activation rather than tumor progression. Additionally, tumor neutrophils exhibit functional diversity: N2-type neutrophils are associated with tumor progression, whereas N1-type populations are characterized by their anti-tumor activities, most notably during immune activation [30]. Hence, the elevation in NLR seen on therapy among our patients could be an indicator signaling a positive shift towards neutrophil activity against tumors.

While our findings identified bNLR as an independent predictor of PFS, several avenues for future research can be explored for a better understanding of the potential roles of the TME and systemic inflammation in cancer progression, and in order to translate these findings into clinical practice. Functional neutrophil characterization using flow-cytometry or single-cell RNA sequencing would verify the polarization towards the N1 phenotype under therapy, validating the positive correlation between an increase in NLR under therapy and PFS we evidentiated in our study and explaining the contradictory findings in the literature. From a clinical perspective, more personalized monitoring strategies and treatment approaches guided by such results should be tested in larger prospective studies before becoming part of generally accepted guidelines.

Several limitations warrant discussion. The study was retrospective in nature and conducted at a single center with a comparatively small sample size, thus limiting the generalizability of our results. Despite excluding patients with active infections, different inflammatory conditions, chemotherapy-induced neutrophilia, treatment-related lymphopenia or medication therapies could have altered NLR levels. Moreover, functional assays to confirm neutrophil phenotypes were not available, leaving the mechanistic hypothesis speculative.

These results highlight the complexity of systemic inflammation in the context of anti-EGFR therapy and underscore the need for larger, prospective studies to validate NLR and its dynamics as practical biomarkers in clinical decision-making.

## 5. Conclusions

Our study demonstrates that bNLR and its early changes can serve as promising prognostic biomarkers for mCRC patients treated with anti-EGFR therapy. A low bNLR was associated with improved PFS. Interestingly, an increase in NLR under treatment was also correlated with better outcomes, suggesting that treatment-induced inflammatory response carries prognostic significance in the context of EGFR inhibition. These findings emphasize the potential of NLR, an inexpensive and widely available biomarker, to complement molecular testing in guiding anti-EGFR therapy.

## Figures and Tables

**Figure 1 jpm-16-00009-f001:**
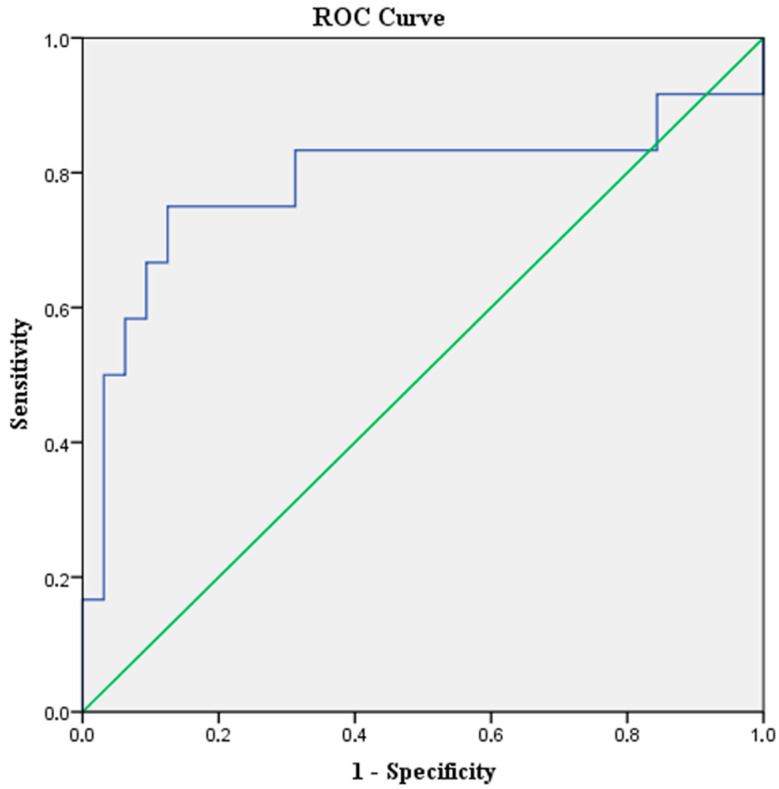
ROC curve analysis for bNLR cut-off value in relation to DCR. The blue line represents the experimental data and the green diagonal line represents the reference (AUC = 0.5).

**Figure 2 jpm-16-00009-f002:**
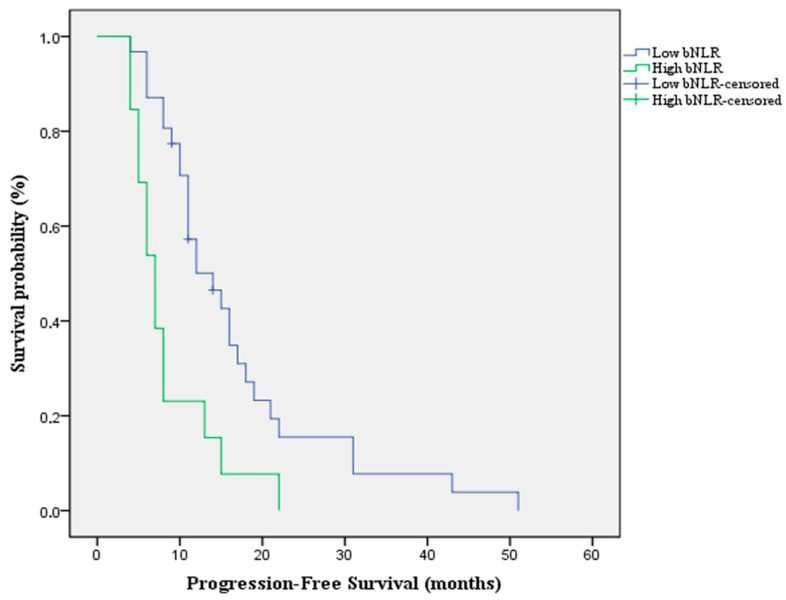
Kaplan-Meier survival curve for PFS according to bNLR that shows an overall better survivability in the low bNLR group compared to the high bNLR group.

**Figure 3 jpm-16-00009-f003:**
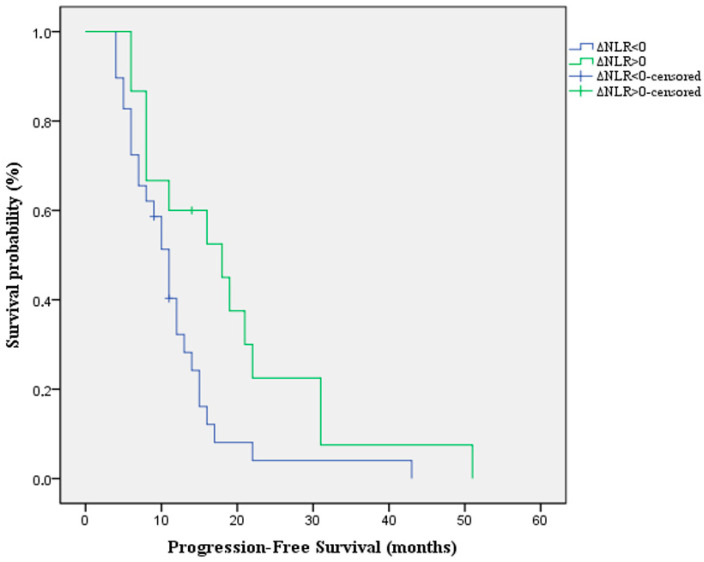
Kaplan-Meier survival curve for PFS according to the alterations of NLR under treatment.

**Table 1 jpm-16-00009-t001:** Patients’ characteristics regarding age, sex, treatment, metastasis, and NLR.

Characteristics		Overall *n* (44)	Percentage (%)
Age at diagnostics, median (range)		59 (24–79)	
Sex	Female	21	48
	Male	23	52
Chemotherapy	FOLFOX	35	80
	FOLFIRI	9	20
Liver metastases	Yes	31	70
	No	13	30
Lung metastases	Yes	12	27
	No	32	73
Peritoneal metastases	Yes	10	22
	No	34	88
Timing of metastasis	Synchronous	24	55
	Metachronous	20	45
bNLR	<3.06	31	70
	≥3.06	13	30
ΔNLR	Negative	29	66
	Positive	15	34

**Table 2 jpm-16-00009-t002:** Univariate and multivariate analysis of PFS, measuring the Hazard Rate (HR) of each characteristic and the *p*-value.

Characteristics	Univariate	*p*-Value	Multivariate	*p*-Value
Age at diagnosis (>59 vs. ≤59)	1.037 (0.556–1.936)	0.909	-	-
Gender (female vs. male)	1.046 (0.556–1.968)	0.889	-	-
CEA value (>24.5 vs. ≤24.5)	1.111 (0.812–1.522)	0.510	-	-
CA 19-9 value (>91.6 vs. ≤91.6)	1.221 (0.866–1.720)	0.254	-	-
Timing of metastasis (synchronous vs. metachronous)	2.107 (1.111–3.995)	0.022	1.803 (0.893–3.639)	0.100
bNLR (>3.06 vs. ≤3.06)	2.711 (1.362–5.398)	0.005	2.282 (1.085–4.801)	0.030
ΔNLR (negative vs. positive)	1.416 (1.034–2.036)	0.031	1.315 (0.870–2.634)	0.194

## Data Availability

The raw data supporting the conclusions of this article will be made available by the authors on request.

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
