# Peer review of "Baseline and Dynamic Neutrophil-to-Lymphocyte Ratio as Prognostic Biomarkers in Metastatic Colorectal Cancer Patients Treated with Panitumumab"

_jpm, 2025, doi:10.3390/jpm16010009_

Round 1
Reviewer 1 Report
Comments and Suggestions for Authors
This paper discusses the value of neutrophil to lymphocyte ratio (NLR) on progression free survival (PFS) in patients with metastatic colorectal cancer (mCRC). The authors need to be commended for their effort in putting together a thoughtful study, however:
(1) The finding that a positive delta LNR was associated with improved PFS needs to be explained further and contextualized with existing literature. The authors should be encouraged to analyze previous papers with similar findings as well as contrast their findings with papers that have shown the opposite.
(2) The authors need to contextualize their findings with existing literature further. The majority of the discussion is focused on facts that although interesting, they may be better suited for the introduction.
(3) Please provide a reference for lines 62-64
(4) The authors need to be encouraged to provide a rationale for the selection of the time points to evaluate the NLRs as well as the cycles of FOLFOX and Folfiri that patients received.
Author Response
This paper discusses the value of neutrophil to lymphocyte ratio (NLR) on progression free survival (PFS) in patients with metastatic colorectal cancer (mCRC). The authors need to be commended for their effort in putting together a thoughtful study, however:
We appreciate the time spent evaluating our manuscript. We are confident that, by responding to reviewer’s comments, we will improve the current paper.
(1) The finding that a positive delta LNR was associated with improved PFS needs to be explained further and contextualized with existing literature. The authors should be encouraged to analyze previous papers with similar findings as well as contrast their findings with papers that have shown the opposite.
We thank the reviewer for this insightful comment. We have modified the Discussion section to further interpret the association between a positive delta NLR and improved PFS. Specifically, we contextualized our findings with previous studies. (lines 222-224 and 227-229).
(2) The authors need to contextualize their findings with existing literature further. The majority of the discussion is focused on facts that although interesting, they may be better suited for the introduction.
We thank the reviewer for this constructive feedback. In response, we have restructured the Introduction and Discussion sections: background facts and general context have been moved to the Introduction, while the Discussion now focuses more on interpreting our findings in the context of existing literature, highlighting similarities, differences, and potential biological explanations.
(3) Please provide a reference for lines 62-64
We thank the reviewer for pointing this out. We have added an appropriate reference to support the statement in lines 62–64. The manuscript has been updated accordingly.
(4) The authors need to be encouraged to provide a rationale for the selection of the time points to evaluate the NLRs as well as the cycles of FOLFOX and Folfiri that patients received.
We thank the reviewer for this comment. The time points for evaluating NLRs were selected to correspond to baseline and after four cycles of therapy, in line with ESMO guidelines recommending a response evaluation every 8–12 weeks for patients with metastatic colorectal cancer. This allowed us to assess early changes in NLR during standard-of-care chemotherapy with FOLFOX or FOLFIRI. The manuscript has been updated in the Methods section (lines 134-136) to clarify this rationale.
Reviewer 2 Report
Comments and Suggestions for Authors
This retrospective study evaluates the prognostic value of baseline neutrophil-to-lymphocyte ratio (bNLR) and early NLR changes (ΔNLR) in 44 patients with left-sided, KRAS wild-type metastatic colorectal cancer treated with first-line chemotherapy plus panitumumab. The authors report that low bNLR (<3.06) is associated with longer progression-free survival (PFS), in line with previous findings. Interestingly, they also observe that an increase in NLR after four treatment cycles correlates with better PFS, which diverges from existing literature.
Major Comments
- The observation that increasing NLR is associated with longer PFS contradicts most prior studies in CRC and other solid tumors, which generally associate rising NLR with poorer prognosis. Alternative explanations (e.g., chemotherapy-induced neutrophilia, treatment-related lymphopenia, selection bias) should be discussed. The biological rationale provided (N1 vs N2 neutrophils) is speculative and needs stronger support from literature.
- The ROC analysis uses DCR as the outcome to derive the bNLR cutoff, while PFS is the main endpoint in survival analyses.
- Important clinical covariates were not included such as number and volume of metastases, and comorbidities or medications influencing NLR
- ΔNLR is dichotomized simply as positive vs negative, but the magnitude and distribution of ΔNLR are not reported. Also the rationale for dichotomization rather than continuous modeling is not provided.
- Only three patients were censored, which is unusually low for a dataset spanning almost 10 years. Please report median follow-up time and clarify how progression was determined during follow-up.
- While patients with active infections were excluded, the manuscript does not describe how “active infection” was determined and whether inflammatory markers (e.g., CRP) were assessed
Minor Comments
- Define PFS, CR, PR, ORR, DCR in the first mention, which is Material and methods and not Results
- How were the cut-off values defined in the univariate and multivariate analysis of PFS for age, CEA, CA 19-9.
Author Response
This retrospective study evaluates the prognostic value of baseline neutrophil-to-lymphocyte ratio (bNLR) and early NLR changes (ΔNLR) in 44 patients with left-sided, KRAS wild-type metastatic colorectal cancer treated with first-line chemotherapy plus panitumumab. The authors report that low bNLR (<3.06) is associated with longer progression-free survival (PFS), in line with previous findings. Interestingly, they also observe that an increase in NLR after four treatment cycles correlates with better PFS, which diverges from existing literature.
We appreciate the time spent evaluating our manuscript. We are confident that, by responding to reviewer’s comments, we will improve the current paper.
Major Comments
- The observation that increasing NLR is associated with longer PFS contradicts most prior studies in CRC and other solid tumors, which generally associate rising NLR with poorer prognosis. Alternative explanations (e.g., chemotherapy-induced neutrophilia, treatment-related lymphopenia, selection bias) should be discussed. The biological rationale provided (N1 vs N2 neutrophils) is speculative and needs stronger support from literature.
The limitations of this study, as mentioned in lines 238-244, that the ratio could be modified due to chemotherapy-induced neutrophilia, treatment-related lymphopenia and also be connected to the relative small number of patients included in the study and the retrospective nature of the study are factors that can influence the results. These are mentioned as limitations in our manuscript.
2. The ROC analysis uses DCR as the outcome to derive the bNLR cutoff, while PFS is the main endpoint in survival analyses.
We admit that the ROC analysis uses the DCR for outcome as seen in the article : Lasagna A, Muzzana M, Ferretti VV, et al. The Role of Pre-treatment Inflammatory Biomarkers in the Prediction of an Early Response to Panitumumab in Metastatic Colorectal Cancer. Cureus. 2022;14(4):e24347. doi:10.7759/cureus.24347 can be considered as a analysis setback. The initial analysis was developed under this circumstances for design. The PFS is considered a surrogate survival endpoint in major clinical trials in Oncology so it can be interpreted in clinical scenarios.
3. Important clinical covariates were not included such as number and volume of metastases, and comorbidities or medications influencing NLR
We agree with the opinion of the reviewer that the number of metastases can alter the results considering that oligometastatic disease can pursue a different treatment schedule. Due to the limited number of data found in the electronical files of the patients about comorbidities and medications used, we couldn’t add this reliable information. We agree that the use of granulocyte stimulating factors could influence these results.
4. ΔNLR is dichotomized simply as positive vs negative, but the magnitude and distribution of ΔNLR are not reported. Also the rationale for dichotomization rather than continuous modeling is not provided.
We thank the reviewer for the comment. We added in the manuscript the following sentence: “The median ΔNLR was -0.466 (range: -6.28 to 1.79).”
5. Only three patients were censored, which is unusually low for a dataset spanning almost 10 years. Please report median follow-up time and clarify how progression was determined during follow-up.
We thank the reviewer for this important comment. The primary endpoint of our study was progression-free survival (PFS). Progression was assessed according to RECIST 1.1 criteria based on CT imaging performed during routine clinical visits, as well as clinical evaluation documented in patient records. Patients were considered censored if they were alive and progression-free at the last follow-up. In our cohort, only three patients were censored, as the majority experienced progression during follow-up. Median follow-up was 11 months.
6. While patients with active infections were excluded, the manuscript does not describe how “active infection” was determined and whether inflammatory markers (e.g., CRP) were assessed
We thank the reviewer for raising this important point and for the opportunity to clarify our methodology. We have revised the manuscript to explicitly define how active infection was assessed. We added in the manuscript the following sentence: “Patients with prescribed antibiotic treatment, modified C-reactive protein, modified procalcitonin or positive for SARS-CoV-2 virus were excluded due to suspicion of active infections.”
Minor Comments
- Define PFS, CR, PR, ORR, DCR in the first mention, which is Material and methods and not Results
We thank the reviewer for this helpful suggestion. All abbreviations, including progression-free survival (PFS), complete response (CR), partial response (PR), objective response rate (ORR), and disease control rate (DCR), have now been defined at their first occurrence in the Materials and Methods section, and the text has been revised accordingly.
2. How were the cut-off values defined in the univariate and multivariate analysis of PFS for age, CEA, CA 19-9.
We thank the reviewer for this comment. The cut-off values for age, CEA, and CA 19-9 used in both univariate and multivariate analyses were defined based on the median values of the study cohort. The laboratory cut-off values were: 24.5 for CEA and 91.6 for CA 19-9.
Round 2
Reviewer 2 Report
Comments and Suggestions for Authors
Authors have answered to my comments.